# Fully Autonomous Real-World Reinforcement Learning with Applications to Mobile Manipulation

**Charles Sun**\*, **Jędrzej Orbik**\*, **Coline Devin, Brian Yang,**
**Abhishek Gupta, Glen Berseth, Sergey Levine**
{charlesjsun,jedrzej.orbik,coline,brianhyang,abhigupta,gberseth,svlevine}@berkeley.edu
**Berkeley AI Research**

**Abstract:** We study how robots can autonomously learn skills that require a combination of navigation and grasping. While reinforcement learning in principle provides for automated robotic skill learning, in practice reinforcement learning in the real world is challenging and often requires extensive instrumentation and supervision. Our aim is to devise a robotic reinforcement learning system for learning navigation and manipulation together, in an *autonomous* way without human intervention, enabling continual learning under realistic assumptions. Our proposed system, ReLMM, can learn continuously on a real-world platform without any environment instrumentation, without human intervention, and without access to privileged information, such as maps, objects positions, or a global view of the environment. Our method employs a modularized policy with components for manipulation and navigation, where manipulation policy uncertainty drives exploration for the navigation controller, and the manipulation module provides rewards for navigation. We evaluate our method on a room cleanup task, where the robot must navigate to and pick up items scattered on the floor. After a grasp curriculum training phase, ReLMM can learn navigation and grasping together fully automatically in around 40 hours of autonomous real-world training.

**Keywords:** Mobile Manipulation, Reinforcement Learning, Reset-Free

## 1   Introduction

Learning-based approaches have the potential to bring robots into open-world environments with end-to-end vision-based policies that can perform tasks without external instrumentation, AR-tagging, or expensive sensors. Such learning systems can be particularly beneficial for mobile manipulators, which perform tasks while navigating through open-world environments that are not practical to instrument or map out in advance. Unfortunately, in practice, most real-world reinforcement learning (RL) systems require careful environment instrumentation and human supervision or demonstrations during the training process to ensure that the robot performs effective exploration and resets between trials, making them dif-

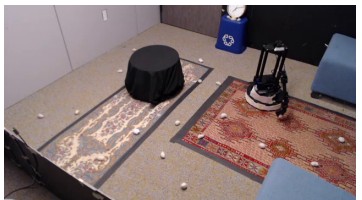

Figure 1: ReLMM enables learning mobile manipulation skills autonomously in the real world, using only on-board sensing.

ficult to apply to the kinds of open-world domains where we might want mobile manipulators to operate. Such systems might require specially installed infrastructure that provides explicit resets [1, 2, 3, 4], or the presence of a person providing resets and monitoring the learning process [5, 6], and cannot simply be dropped into a natural environment and continue learning.

To address this issue and make it possible for mobile manipulators to learn with RL directly in the real world, we propose a system for learning mobile manipulation skills without instrumentation, demonstrations, or manually-provided reset mechanisms. We aim to produce a system that enables a robot to learn autonomously in settings such as homes and offices, such that anyone could simply

5th Conference on Robot Learning (CoRL 2021), London, UK.

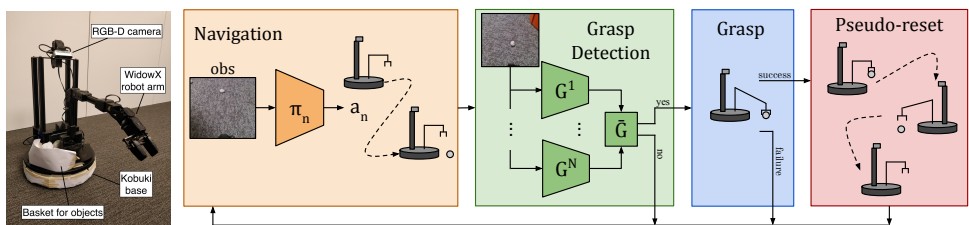

Figure 2: **Method overview.** ReLMM partitions the mobile manipulator into a navigation policy and grasping policy. Both policies are rewarded when an object is grasped. We use an ensemble of action-conditioned grasp success prediction functions estimate the success of a potential grasp better and use the uncertainty as an exploration bonus. If grasp success is likely, a grasp action is sampled from the grasp predictors and executed. If the grasp is successful during training, the robot executes a pseudo-reset by placing the collected object back down in a random location.

place the robot down, start the learning process, and return to a trained robot. This goal dictates several constraints that shape our method: (1) the robot must learn entirely from its own sensors, both to select actions and to compute rewards; (2) the entire learning process must be efficient enough for real-world training; (3) the robot must be able to continually gathering data at scale without human effort. A system that meets these requirements would not only be able to learn skills in open-world settings, but could also continue to improve throughout its lifetime: when learning can be performed practically without instrumentation, there is no reason to stop the learning process at deployment time, and the robot keep getting better and better at its given task perpetually. Our aim is not to propose the best possible system for solving any particular task. Rather, we aim to show how to create a real-world reinforcement learning system that enables learning mobile manipulation skills entirely from real-world interaction, with minimal human intervention.

Our contribution is a system for autonomously training a mobile manipulation robot that satisfies the above constraints, which we call Reinforcement Learning for Mobile Manipulation (ReLMM). We apply ReLMM to the task of collecting items scattered across a room. Our system learns directly from on-board ego-centric camera observations and uses proprioceptive grasp sensing to assign itself rewards. To ensure the learning process is sample efficient and to facilitate exploration, we split the robot's controller into separate navigation and grasping neural network policy modules that choose when to act based on their predicted value and are continuously trained together for the same objective: successfully grasping objects in the environment. Separating the policies enables the use of uncertainty-based exploration for the grasping module, which uses an ensemble of Q-functions to explore grasp actions efficiently. Lastly, to reduce the need for humans to provide interventions in the form of resets, we develop an autonomous resetting behavior where the robot re-arranges the environment as it learns, so as to continually create new arrangements of objects for the agent continually "practice." Together, these components plus a brief grasping curriculum enable a system that can operate in real-world environments, learning how to navigate and grasp from its own collected experience, and mastering room cleanup tasks with about 40-60 hours of autonomous interaction. Videos are available at `https://sites.google.com/view/relmm`

## 2    Related Work

Robotic mobile manipulation tasks pose a number of unique challenges [7]. Many prior methods have addressed these challenges by requiring human effort for instrumentation and state estimation [8, 9, 10, 11], hand-coded controllers [12], or demonstrations [10, 13, 14]. Several methods also require external instrumentation such as top-down camera views, oracle knowledge of object pose, or precomputed navigation maps [15, 16, 17]. textcolorMobile manipulation has also seen benefits from combining learning and planning for more efficient exploration in simulation [18, 19, 20], which allow for safe and effective behavior in the real world if object or goal locations are provided. While these choices are pragmatic, they do not address the problem of learning continually in uninstrumented real-world settings and require a simulated world to train in. In contrast, our proposed system is aimed at enabling reinforcement learning directly on the real hardware that is maximally autonomous, and does not require external instrumentation.

Hierarchical reinforcement learning has been shown to learn interactive navigation and object re-arrangement tasks in simulation operating from first-person view [21, 22], but require millions of

timesteps to train and have not been demonstrated in the real world. Our system uses a similar hierarchical structure to these, but targets accessible real world learning by leveraging policy uncertainty estimates and curriculum learning for realistic sample efficiency.

While RL-based methods allow agents to improve via interaction, enabling robots that can learn outside of instrumented laboratory settings is difficult. Such settings lack episodic structure and well-shaped reward functions [23, 24, 25, 26, 27] that are important for success [28, 29]. Large-scale uninterrupted deployment and training in the real world has been studied in several prior works [30, 31, 32, 33, 34, 35, 36]. While many of these works train robots in the real world without human interventions, they focus on navigation without manipulation. Large-scale real-world learning has also been successful for robotic grasping in laboratory settings. By letting robots learn from their own experience, prior approaches have shown that robots can learn to grasp from images [37, 38, 39, 40] or point clouds [41, 42]. These methods show real world learning can lead to robust robotic behavior, but they do not tackle challenging mobile manipulation tasks.

Gupta et al. show an approach to training a grasping policy on a mobile manipulator, but unlike our work, do not attempt to learn to navigate [43]. Wang et al. train a mobile manipulator with RL in simulation, but require known object pose, dense rewards, and episodic resets to a single initial state [16]. In contrast, our approach learns entirely from a sparse reward and onbaord sensors. Past approaches to learning without episodic resets have focused on stationary manipulation or simulation [44, 45, 2]. In this work we explicitly focus on how *mobile manipulation* robots can learn without external sources of resets or state estimation in the real world and lay out a set of design decisions that makes this process a practical one for acquiring robot skills.

## 3 Preliminaries

In this work, we use reinforcement learning (RL) as a general purpose algorithm for learning robotic behaviors. Reinforcement learning has the advantage of being able to operate on autonomously collected data, and enables the robot to improve through trial and error. To this end, the mobile manipulation task is formulated as a partially observed Markov decision processes (POMDP) with an observation space $\mathcal{O}$ of first person RGB images, state space $\mathcal{S}$, action space $\mathcal{A}$, reward function $r(s_t, a_t)$, transition dynamics $\mathcal{P}(s_{t+1}|s_t, a_t)$, observation probability $\mathcal{P}(o_t|s_t)$, a discount factor $\gamma$, and an initial state distribution $\rho(s_0)$. The goal of reinforcement learning is to learn a control policy $\pi(a_t|o_t)$ that can determine which actions to take in each observation such that the expected sum of rewards is maximized. This objective can be written as $J(\pi) = \mathbb{E}_{a_t \sim \pi(o_t), s_t, o_t \sim \mathcal{P}} \left[ \sum_t \gamma^t r(s_t, a_t) \right]$, as for a standard RL problem.

## 4 ReLMM: RL For Mobile Manipulation

We develop the ReLMM system to enable training mobile manipulation robots with RL. While we specifically apply it to a room cleaning task, in principle ReLMM could be used to learn other mobile manipulation tasks as well. Each component of ReLMM is chosen in order to maximise the autonomy of learning while retaining the sample efficiency needed to train in the real world. The specific task that we study in our experiments involves training a robot to quickly navigate around in a room with obstacles, physically pick up many objects, and place them in a basket mounted on the robot, as shown in Figure 1 and 3.

---

**Algorithm 1** TrainGrasp($G^1, ..., G^M, \mathcal{D}_g, N, st$)

1: **for** t = 0, ..., $N$ steps **do**
2:   Get grasp observation $\tilde{o}$.
3:   Sample $a_g \sim \pi_g(\cdot|\tilde{o})$. // see Equation 2
4:   Perform grasp $a_g$, receiving $r_g = 0$ or 1.
5:   Store $(\tilde{o}, a_g, r_g)$ in $\mathcal{D}_g$.
6:   Update $G^1, ..., G^M$ on $\mathcal{D}_g$
7:   **if** $st$ =True, Drop object if held.
8:   **elseif** $r_g = 1$ and $st$ =False, **return** $r_g$
9: **return** 0

---

Our system provides for efficient autonomous learning by decomposing the policy into grasping and navigation policies, using an ensemble of grasping models to explore based on uncertainty, automatically rearranging the environment after successful grasps, and using a curriculum to bootstrap and stabilize the concurrent training of both policies. Our final system can learn room cleaning skills in a number of different room configurations in $\sim$ 40 hours directly in the real world.

## 4.1 Grasping Policy Training

As noted previously, we decomposed the control problem into grasping and navigation. This gives the manipulation policy two objectives: given an image observation, accurately model the robots chance of success should it attempt a grasp and, if so, select an appropriate action to maximize success. We obtain the former by training an ensemble of grasping policies and using their uncertainty to efficiently explore grasping. For choosing *how* to grasp, the policy must learn with sufficiently low sample complexity so as to make real-world training feasible. To reduce the complexity of the exploration problem we formulate grasping as a discrete single-step top-down action. The grasp policy $\pi_g(a_g|o_g)$ is parameterized with the action $a_g$ discretized in the x-y plane, and the observation $o_g$ corresponding to a RGB image from the robot's camera. Such single-step action selection formulations are amenable to more efficient training than more complex multi-step tasks [46, 47].

Framed in this way, the grasping task corresponds to a contextual multi-armed bandit problem. Specifically, we train grasping policies that, given an image, predict the likelihood of grasp success for each action.We use a soft-max over the action values to sample actions in proportion to their exponentiated probability of success. To create the ensemble, we train $M = 6$ independent grasp policies, $G^1 \ldots G^M$ that are each by minimizing the cross-entropy loss on the same dataset:

$$\mathcal{L}_g^i = \mathbb{E}_{(o_g, a_g, r_g) \sim \mathcal{D}_g}[-r_g \log G^i(o_g, a_g) - (1 - r_g) \log(1 - G^i(o_g, a_g))]. \tag{1}$$

Here, $r_g$ is 1 when the robot successfully grasps an object, which is determined by presenting the gripper to the onboard camera. The grasping exploration policy is formed by constructing a Boltzmann distribution from optimistic estimates of grasp success, where the mean estimate from the ensemble, $\mathbb{E}[G^i(o_g, a_g)]$, is modified by adding a multiple of the ensemble variance $\sigma(G^i(o_g, a_g))$, which we expect to be larger for actions where success is more uncertain:

$$\tilde{G}(o_g, a_g) = \alpha \mathbb{E}[G^i(o_g, a_g)] + \beta \sigma(G^i(o_g, a_g)), \tag{2}$$

with $\pi_g(a_g|o_g) \propto \exp(\tilde{G}(o_g, a_g))$. The expectation and standard deviation are taken over the ensemble, and $\alpha, \beta \geq 0$ are hyperparameters. Other equivalent multi-step off-policy or contextual bandit algorithms can be used to train the grasping policy. However, we show in Section 6 that they are not as sample efficient for the mobile manipulation task in this work. Algorithm 1 describes the grasp training process in further detail.

## 4.2 Navigation Policy Training

For navigation, the policy must be able to control the mobile base to approach objects in a way that the current grasping policy can succeed. The navigation policy $\pi_n(a_n|o)$ outputs the action $a_n$ that controls the forward and turn velocities of the mobile robot base.

At every time step, the agent has to decide whether to perform a grasp or not by balancing the opportunity of receiving reward, the chance to collect novel data for the grasping policies, and the cost of wasting a timestep if there is no object within reach. This balancing act is done by reusing the same uncertainty measure described in Equation 2 when choosing whether to attempt a grasp. The probability of whether to attempt a grasp is

$$\mathbb{P}[\text{grasp}|o] = \max_{a_g} \tilde{G}(o, a_g). \tag{3}$$

Under this design, the navigation policy $\pi_n(a_n|o)$ continues to experience observations and output navigation actions, and at every step the choice of whether to perform a grasp or not is made by sampling from grasp success probability of the current grasp model $\mathbb{P}[\text{grasp}|o]$ defined in Eqn 3. When the model decides attempting a grasp is worth the risk, the robot executes a grasp and evaluates the outcome to provide the navigation agent a reward. From the perspective of the navigation policy, the choice of whether to grasp or not is a part of the inherent dynamics of the environment. We compare two possible rewards $r_n$ for the the navigation policy. The first option is directly optimizing for the task by rewarding the navigation when a grasp is successful (i.e. where the current grasp ensemble has high performance): $r_n(o) = r_g(o) - 1$. The second option is to reward the navigation for reaching states that the current grasp ensemble will choose to grasp at, which is a function of its mean and uncertainty: $r_n(o) = (\mathbb{P}[\text{grasp}|o] - 1)$ that we use to *relabel* states without successful grasps during SAC policy update. As this reward function only depends on the grasp ensemble and not on actual grasp success, this reward is computed at every policy update step and

in Figure 5 we show how it can improve sample efficiency. The RL objective for navigation is $\max_{\pi_n} \mathbb{E}_{\pi_n(a_n|o_t)} \left[ \sum_{k=0}^{\infty} \gamma^k r_n(o_{t+k}) \right]$ We train the navigation policy for this objective using soft actor critic (SAC) [48, 49].

## 4.3 Training with Autonomous Pseudo-Resets

While the training schemes described above allow ReLMM to learn efficiently, both the contextual bandit grasping formulation and the navigation training setup requires an episodic training setup, where the environment is reset between trials, for example by replacing the objects into the world at random positions. To enable the robot to learn mobile manipulation skills without manually provided resets, we construct an automated pseudo-reset system that allows our method to learn autonomously without human intervention. After a successful grasp, the environment would ideally be reset by relocating the object to a new, randomly selected location. In stationary bin grasping setups, this can be automated simply by dropping the object back into the bin. However, for a mobile manipulator, dropping the object back to where it was grasped would promote policies that fail to search for new objects, and simply remain in the same location. To force the

---

**Algorithm 2** ReLMM (with Stationary Curriculum)

1: Hyperparameters: $M$, $N_{\text{st}}$, $N_{\text{grasp}}$, *relabel*
2: Init: function estimators $\pi_n$, $G^1, \ldots, G^M$.
3: Replay buffers $\mathcal{D}_n = \{\}$, $\mathcal{D}_g = \{\}$
4: TrainGrasp($G^1, .., G^M, D_g, N_{\text{st}}$, True)
5: **for** $t = 0, \ldots, T$ steps **do**
6:   Get navigation observation $o_t$
7:   Sample $a_n \sim \pi_n(\cdot|o_t)$ and perform $a_n$
8:   **if** *uniform*$() \leq \mathbb{P}[\text{grasp}|o_t]$ **then**
9:     $r_g =$TrainGrasp($G^1, .., G^M, D_g, N_{\text{grasp}}$,False)
10:   **else** $r_g = 0$
11:   **if** *relabel* **then**
12:     Navigation reward $r_n = \mathbb{P}[\text{grasp}|o_t] - 1$
13:   **else** $r_n = r_g - 1$
14:   Get next navigation observation $o_{t+1}$
15:   Store $(o_t, a_n, r_n, o_{t+1})$ in $\mathcal{D}_n$.
16:   Update $\pi_n$ with $\mathcal{D}_n$ using SAC.
17:   Pseudo-reset
18: **end for**

---

policies to learn to grasp in varied situations, we perform an automated random pseudo-reset, by commanding random navigation actions while the robot is holding the object, placing down the object in this new location, and then navigating randomly away. This ensures the robot will not always be near an object during training and must instead learn to seek objects out. We outline the overall algorithm used for training in Algorithm 2.

## 4.4 Training Curricula

The separate grasping and navigation policies lend themselves naturally to curricular training, where the grasping policy, which needs to have some successes to provide rewards to the navigation policy, can be prioritized at the beginning of the learning process. We propose two types of curricula, which both break the potential "chicken-and-egg" training problem of providing poor reward signals for navigation from an untrained grasping model.

**Stationary curriculum.** The simplest curriculum is to place a single object in front of the robot and run Algorithm 1 with $st = $ True for $N_{\text{st}} = 2000$ steps. After each successful grasp, the robot places the object down randomly. If the object is knocked into an area the robot can not reach (because the base is stationary), which happens about $5\%$ of the time, a human observer must push the object back into the graspable area. This curriculum is very time efficient, but does require occasional human intervention.

**Autonomous curriculum.** For fully autonomous learning, we develop a training curriculum that favors collecting grasping data early on by performing a high number of grasps at the beginning of training, according to hyperparameters $N_{start}$, $N_{stop}$, and $N_{max}$. The robot attempts $N_{grasp} = N_{start}$ grasps after every navigation step. If the robot succeeds at grasping an object it will practice with this object until $N_{grasp} = N_{stop}$ unsuccessful grasps. This initial automatic grasping curricula ends when a total of $N_{max}$ grasps are complete. More details on the grasping curriculum algorithm are in Appendix **??**.

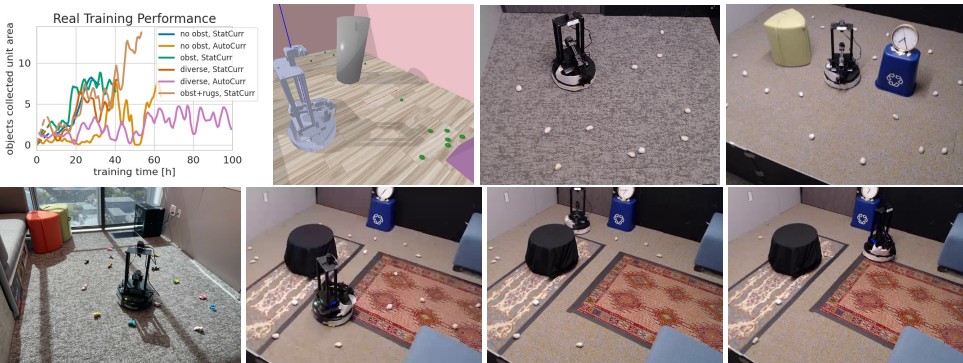

Figure 3: **Top Left**: Learning curves for training the real robot showing the number of objects grasping in the last 15 min of operation, which is $\sim 250$ actions. This metric are divided by the rooms' surface areas to make them comparable. **Top mid-left**: The simulated environment, which we run with and without obstacles. **Right & Bottom**: Snapshots from the evaluation tasks without obstacles in a $4m^2$ room (top-mid-right, *no obst*, bottom-left, *diverse*), with obstacles *obst* in a $9m^2$ room (top-right), and with obstacles and rugs *obst+rugs* in a $10m^2$ room (bottom-right). Videos are available at `https://sites.google.com/view/relmm/home`

## 5 Robotic Mobile Manipulator: System Overview

Our choice of robotic system reflects the need for a robust robotic platform that can operate autonomously for long periods of time, and is unlikely to cause damage to itself and its surroundings. Ensuring safety during autonomous operation is itself a significant research challenge, which is outside of the scope of this work. Therefore, we utilize a small-scale low-cost mobile manipulation platform based on the LocoBot design [43, 50], shown in Figure 2 (left), which consists of an iClebo Kobuki mobile base and a WidowX200 5-DoF arm. The robot sensors include an Intel RealSense D435 camera at the top of the robot, as well as bump sensors on the base. We use an onboard Intel NUC to command the robot, and connect wirelessly to a server for data processing and training. Random exploration is generally safe with the LoCoBot as it is small, light, and will automatically stop the arm's motors if they encounter resistance. We also use the depth camera output to automatically avoid collisions, as described in **??**. The robot learns in a real-world office space, with varied lighting conditions, distractors, and surface textures. Our experiments utilize small objects that the robot can feasibly pick up, which consist of socks and toys – a sampling of objects one may want a robot to pick up off the floor.

To control the robot, we separately command the iClebo mobile base and the WidowX200 robotic arm with the corresponding navigation and a grasping policies. The navigation policy stops the robot during a grasp. For grasping, we use a directed end-effector control space. Assuming the floor is flat, the vertical position for grasping is always chosen to be just above the floor, and the learning algorithm chooses a point in $X - Y$ space in front of the robot to perform a grasp at. This chosen position then dictates where to move the gripper using inverse kinematics. Details on the learned network architectures are in Appendix **??**.

## 6 Experimental Results

Our experiments aim to evaluate our autonomous reinforcement learning system in a number of real-world environments, as well as to provide ablation experiments and analysis in simulation. In particular, we aim to study the following questions: (1) Can ReLMM learn autonomously in the real world? (2) How does the control hierarchy affect learning performance? (3) How does ReLMM compare to other policy designs and prior methods? Our goal is to study real-world reinforcement learning, rather than necessarily provide the best possible solution to the room clearing task, and hence we design our system to be general, with only the reward determining the task. **Experiment details** For all real-world experiments, the entire training procedure is performed in the real world on the LoCoBot platform, with about 25 to 50 hours of training depending on the environment. The training is autonomous, with the exception of the stationary curriculum (if used), and battery changes every $\sim 5$ hours, during which time we may replace objects that become stuck near corners, or walls.

As our task is non-episodic, the policies are evaluated at the end of training. This is done by scattering the objects, executing the policy (without pseudo-resets) for 15 minutes, and counting the number of objects collected in that time. We use four real environments in our evaluation (no obstacles, with obstacles, with diverse objects, and with obstacles and rugs and a simulated environments, as shown in Figure 3. Each room has a different size and number of objects, so we report the percent of objects collected in each 15 minute block of training, as shown in Figure 3. In simulation, we plot the evaluation performance using 250 timesteps instead of 15 minutes.

We compare our approach to several prior methods and baselines. The baselines include: **Rand all**, a policy that selects navigation and manipulation actions randomly, as a lower baseline; **Rand nav**, a method similar to Gupta et al. [43] that collects grasping data in the same way as our method, but has a random navigation controller. These baselines disentangle the benefits we get from learning both the grasping and the navigation. We also include a hand-coded controller (**Scripted**) specifically engineered for this task that locates objects by thresholding the image pixels and grasps at their centroids, which provides a strong hand-designed baseline. For more detail on the implementation of these baselines, see Appendix 6.

**Real Robot Evaluation** Our real-world experiments evaluate how well ReLMM can learn a mobile grasping policy autonomously in a variety of rooms. We conducted separate experiments for each of the rooms shown in Figure 3, which differ in terms of size, furniture, layout, and objects. ReLMM can train using both the *stationary* and *autonomous* grasping curricula, with the *autonomous curriculum* requiring less human effort, at the cost of increased training time. The result are summarized in Table 1 with the training time given in **??**. Examples of the learned behaviors are also shown in **??**, and are illustrated in more detail in the supplementary video.

| Env | no obst | obst | diverse | obst+rugs |
|---|---|---|---|---|
| *ReLMM-StatCurr* | **88 ± 2** | **93 ± 2** | 63 ± 8 | **78 ± 6** |
| *ReLMM-AutoCurr* | 77 ± 8 | – | **75 ± 4** | – |
| *Scripted* | 75 ± 4 | 88 ± 6 | 56 ± 3 | 65 ± 9 |
| *Rand nav* [43] | 52 ± 12 | 38 ± 10 | 22 ± 9 | 20 ± 7 |
| *Rand all* | 12 ± 6 | 2 ± 2 | 2 ± 3 | 5 ± 4 |

Table 1: Percentage of objects that the robot collects during eval in each environment (shown in Figure **??**) (higher is better). Each method is trained once per env, and evaluated 3 times. The numbers are mean and stddev of the 3 evaluations. ReLMM outperforms the baselines by learning both grasping and navigation jointly, without requiring environment instrumentation. Due to ReLMM-AutoCurr's slower learning, we only evaluate it in *no obst* with *diverse*. In **??** we provide the training time in each env.

To provide context for these results, we compare ReLMM to the previously described baselines. The *Rand all* and *Rand nav* [43] baselines perform poorly, since neither method learns a directed navigation behavior, making it difficult to effectively navigate the room to collect objects, though *Rand nav* [43] is effective at picking up the objects if it stumbles upon them. This illustrates the importance of an intelligent navigational strategy in these settings. The *Scripted* baseline, which was specifically engineered for this task, performs reasonably in most settings, but still falls short of the policy learned by ReLMM in all of the environments, and crucially cannot improve from these results autonomously. This is particularly important in the *diverse* and *obst+rugs* environments where it is difficult to find a pixel threshold that works for all objects and all backgrounds, ReLMM can automatically learn how to identify objects to grasp via interaction. In Figure 4, we plot the evaluation performance of our ReLMM-StatCurr agent learning in *obst+rugs* for a few checkpoints, showing that our method is improving from its experience and is still improving after 50 hours of training. This is important in open-world settings, where our approach would enable mobile manipulators to improve perpetually over the lifetime of their deployment.

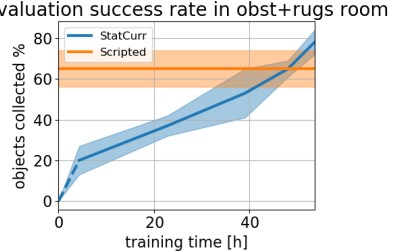

Figure 4: Different checkpoints of ReLMM in the obst+rugs room. Performance improves steadily with more training, and indeed is still increasing even at our final checkpoint, suggesting that lifelong training enables continual improvement for ReLMM.

**Simulation Analysis** To study questions (2) and (3), we perform a detailed ablation analysis in simulation. As discussed in **??**, we find that a discretized grasping policy learns with $\frac{1}{3}$ the samples compared to a continuous one. In Figure 5, we show the performance of various ablations of our

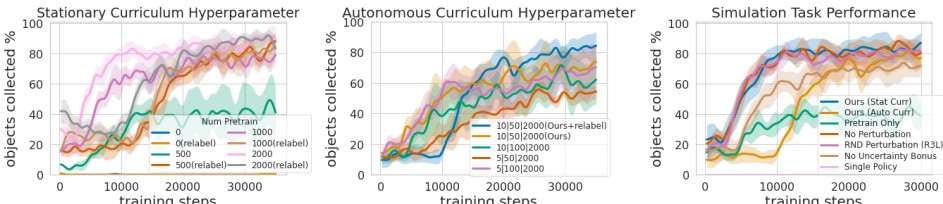

Figure 5: **Performance in the simulated environment without obstacles.** Each plot is the mean and stddev over 3 random seeds. **Left**: Ablation of different number of grasp "pretraining" samples for the stationary curriculum ($N_{st}$) and navigation reward relabeling (described in subsection 4.2) in simulation. Increased pretraining of the grasping policy improves overall mobile manipulation performance. **Center**: Analysis to find the best parameters for the *autonomous* curriculum. With the proper settings ($N_{start}|N_{stop}|N_{max}$) the *autonomous* curriculum can be almost as efficient as the *stationary*, without requiring as much manual effort. **Right**: Ablation of ReLMM, all use the stationary curriculum and no relabeling except for *Ours (AutoCurr)*. We find that the uncertainty bonus and joint training are critical components of our system.

system in the simulated environment with the *stationary curriculum*. First, we ablate the policy decomposition by training a *single policy* with reinforcement learning using grasp success as the reward. The joint action space poses a much larger exploration problem, and the policy is unable to make headway on the task in a reasonable number of samples. This indicates that the hierarchical decomposition used by ReLMM significantly improves training performance. Next we see that freezing the grasp policy after the stationary phase (*Pretrain Online*) and only training the navigation is much worse than training both together. This illustrates the interplay between the two policies, as the navigation is dependent on the grasping for obtaining rewards. As shown in the plot, training the grasp policy without the uncertainty bonus (i.e. $\beta = 0$) leads to significantly poorer performance for ReLMM, as it is less incentivized to explore. Finally, we see that the autonomous curriculum with reward relabeling can get similar final performance as the stationary curriculum with less human intervention at the cost of longer training time.

Lastly, we examine how to get the best performance in terms of sample efficiency by performing an ablation on the choices of curriculum to minimize the samples needed. This is especially important as ReLMM trains two policies concurrently, which can often be unstable. As can be seen in Figure 5 (left) a strong final navigation and grasping policy can be learned with just 500 stationary grasps while applying *relabeling* to the navigation rewards. Next, we expand on the curriculum results on the real robot in order to tune the *autonomous* curriculum to be nearly as sample efficient as the *stationary* curriculum. In Figure 5 (middle) we can see that the best final performance is achieved by increasing the frequency of grasps early on in training and adding a bonus using the grasp model uncertainty.

# 7 Discussion and Future Work

We presented ReLMM, a system for autonomously learning mobile manipulation skills in the real world, without instrumentation, and with minimal human intervention. Our real-world experiments, conducted in four separate rooms, show that ReLMM can train continuously for several days (40-60 hours) with only occasional interventions, and that the resulting policies are effective at cleaning up the room. Furthermore, our experiments show that ReLMM continues to improve with more training, suggesting that it provides an effective approach for lifelong learning for robotic systems deployed in open-world settings. Our simulated ablation studies further provide support for the design decisions in ReLMM, indicating that the hierarchical decomposition of navigation and grasping greatly improves learning performance, autonomous practicing with a suitable exploration bonus enhances learning speed, and our automated curriculum can provide effective performance when learning from scratch. Even when using the manually-provided curriculum, the grasp pretraining phase does not need to be long: only 1000 attempted grasps were needed to get reasonable room-cleaning performance, although more pretraining can improve performance further. Extending this system to more complex manipulation tasks would be an exciting direction for future work. We hope that this system for practical RL on mobile manipulators will make mobile manipulation more accessible to outside lab spaces and to users who need maximally autonomous learning algorithms.

**Acknowledgments**

We thank the anonymous reviewers for their helpful feedback in revising our manuscript. This research was supported by the Office of Naval Research, ARL DCIST CRA W911NF-17-2-0181, the National Science Foundation, and Schmidt Futures.

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
