# OpenReview forum: "Fully Autonomous Real-World Reinforcement Learning with Applications to Mobile Manipulation"
_robot-learning.org/CoRL/2021/Conference — CoRL2021 Poster_

### Official Review · Reviewer_aRW7 · 2021-07-23

**Originality:** Good
**Technical Quality:** Very Good
**Clarity Of Presentation:** Fair
**Impact:** 2

**Recommendation:**

Weak Accept: I recommend accepting the paper, but will not argue for my recommendation if the majority of other reviewers have a different opinion.

**Summary:**

This paper presents a deep reinforcement learning approach for training an agent to both navigate and grasp objects in the real world given only an RGB video feed.

The key idea is to decompose grasping and navigation into two separate modules, with manipulation being trained to estimate the uncertainty of grasp success with rewards given for successful grasps, and navigation being trained to navigate in such a way as to minimize this uncertainty.

Two forms of curriculum are also suggested: (1) the robot is stationary and only does grasping actions, with some human intervention to ensure objects are present, and (2) the robot both navigates and grasps, but performs grasping actions more frequently than navigation actions. To continuously learn in the real world without human intervention, a pseudo-reset mechanism is employed in which the robot randomly moves and releases a grasped object and moves to a random location. Compared to prior work, this overall approach allows for training directly in real world without any input information except an RGB observation stream, which can be done in the span of dozens of hours.

Evaluation for training in the real world shows the approach to be able to achieve high success rates. Several baselines that do not use different grasping and navigation modules or a curriculum are also evaluated, and attain low success rate. Evaluation in simulation also presents ablation results that justify the need for the various components of the system.

**Issues:**

* Cite HRL4IN and compare your approach to their in text. Ideally, also compare your approach to a policy similar to theirs experimentally
* Explain the experimental protocols in greater detail,
* Provide more results for the real world experiments, in particular by presenting full results for Figure 3 and Table 1

Update after responses:
As stated in my comments below, I believe the writing needs to be updated to more strongly reflect the limitations of the contribution in terms of lack of human intervention and the degree to which the system design as opposed to task design allows for efficient learning.

**Reviewer Expertise:**

Excellent: Expert knowledge on the topic of the paper

**Strengths And Weaknesses:**

Strengths:
* The problem of training both navigation and manipulation actions using only raw sensory inputs and a reward function is useful to address, with deep RL only having been explored in this setting in a few prior works
* The paper is clearly written and has few ambiguities, and is easy to follow
* The claim that this approach requires less knowledge about the environment and less human intervention is accurate and significant
* The key new ideas are well suited to the task, and are mostly novel. In particular, using the grasp uncertainty to reward good navigation is a good idea. See caveat below.
* Experiments are partially well done, with real world results presenting impressive success rates and simulated results presenting more detailed justification of the need for parts of the approach. However, see caveats below.
* Compared to prior work, demonstrating learning results entirely from real world experience is impressive and indicates the approach's usefulness.

Weaknesses:
* While the related work surveyed is fairly comprehensive, the work "HRL4IN: Hierarchical Reinforcement Learning for Interactive Navigation with Mobile Manipulators" is not cited. This is problematic, as this work has a similar concept of decomposing the policy into manipulation and navigation components, and training primarily from raw sensor inputs. Its manipulation and navigation policies can arguably be change to this paper's action space and the curriculums suggested could be applied to its approach, meaning that there is significant overlap conceptually. The This work does use additional inputs to the agent such as door locations, but this is not strictly required.
* Additional citations for hierarchical RL approaches would also make sense, given that they also decompose the skills needed to learn (even if not specifically for the task of mobile manipulation).
* Unlike prior work on deep RL for mobile manipulation, this approach does not allow for learning arbitrary mobile manipulations skills, and is instead specialized to navigation and grasping
* The approach is evaluated in a simplified setting of having simple ball-like objects scattered on the floor, in a space of limited size that does not require much navigation. This makes it so both the perception and manipulation aspects are significantly simplified compared to prior tasks in deep RL for mobile manipulation. It is unclear whether this approach can scale to more complex settings.
* The evaluation in the real world is not presented very clearly, with Figure 3 having only partial learning curves and Table 1 having missing entries. This is especially problematic for Figure 3, since the shown curves are quite chaotic. Additionally, variance values are presented in Table 1, and not in Figure 3. These things combined make me worry that the results are partially cherry-picked.
* The baselines being compared to are very simple, with HRL4IN being a much more suitable algorithm to compare against.

**Summary Of Recommendation:**

I recommend rejection, despite this presenting ideas with good originality, technical quality, and potential impact. This is primarily due to issues I presented with respect to the experiments section, and due to weak baselines that do not include an approach similar to HRL4IN, which is also not cited and not compared to (which reflects my evaluation of clarity). However, I hope the author response and revisions could be sufficient to address these issues, in which case I would likely recommend acceptance.

Update after responses:
I have updated my recommendation to weak accept from weak reject - see my reply to the authors in comments below.

---

> ### Author Response · Authors · 2021-08-27
> **Review resposne**
>
> Thank you for your thorough review, we will try to address your concerns to improve the paper.
>
> Q1: “Unlike prior work on deep RL for mobile manipulation, this approach does not allow for learning arbitrary mobile manipulations skills, and is instead specialized to navigation and grasping”
>
> A1: Our aim is to study how fully autonomous real-world reinforcement learning systems could be instantiated. While it's true that different tasks might require a different reward function or a different representation for the manipulation skill, the system building exercise and resulting empirical observation, which shows for the first time that a mobile manipulator can train entirely autonomously for 25-50 hours in the real world and learn a task that requires both manipulation and navigation, provides a recipe that can serve as a starting point for future researchers aiming to study real-world reinforcement learning systems. We believe that this is highly relevant for the CoRL community.
>
> Q2: “The approach is evaluated in a simplified setting of having simple ball-like objects scattered on the floor in a space of limited size that does not require much navigation. This makes it so both the perception and manipulation aspects are significantly simplified compared to prior tasks in deep RL for mobile manipulation.”
>
> A2: We would like to clarify that our aim is not to propose the best possible system for solving this particular task. Rather, we aim to show how to create a real-world reinforcement learning system that enables learning mobile manipulation skills entirely from real-world interaction, with minimal human intervention. We believe that this is of interest to the CoRL community, which focuses on robotic learning and strongly encourages reporting real robot experiments. If there are any prior works on deep RL for mobile manipulation that learn autonomously in the real world and can perform more complex tasks, we would be happy to cite and discuss them, but we are not aware of any such works. To our knowledge, our system is the first to enable a real-world robot to learn a mobile manipulation skill entirely through autonomous real-world training without human intervention. We have edited the introduction to make the scope of our work more clear.
>
> We would also like to clarify that the perception and manipulation aspects of our environments are not trivial:  due to windows, lamps, and shadows from the furniture, the appearance of the room changes significantly over the course of training as the sun rises and sets, which is a challenge unique to real world training. The “diverse” room also tests the method’s ability to learn to collect objects of different shapes and appearances.
>
>
> Q3: “The evaluation in the real world is not presented very clearly, with Figure 3 having only partial learning curves and Table 1 having missing entries. This is especially problematic for Figure 3, since the shown curves are quite chaotic. Additionally, variance values are presented in Table 1, and not in Figure 3. These things combined make me worry that the results are partially cherry-picked.”
>
> A3:
>
> - Figure 3 shows the real robot training performance. While the robot is training autonomously we only check on it every 4-5 hours to replace the battery. As well, some tasks are easier than others and do not need as much training time on the robot. This is why some learning curves are different lengths. Also, due to the desired autonomous learning, we cannot run the evaluation protocol regularly during real-world training (as this would require rearranging the objects which would reset the environment and make the learning more dependent on human interaction). As a substitute, we plot the training performance as the number of successful grasps every 15 minutes. These plots are all noisy because the environment changes over the course of training: the lighting changes and the objects move as the robot grasps them, places them, and bumps into them.
> - The simulation experiments are each performed with 3 random seeds, the mean and standard deviation are plotted in Figure 4. Each real training experiment is only run once per room, but evaluated 3 times, with the mean and standard deviation of the evaluations listed in the table. We have clarified this in the text.
> - The results are not cherry picked, they are simply the performance we got at the end of training. In fact, we were also able to continue training the agents in the obstacle and obstacle+rugs rooms for 12 hours each, and found this increased their performance by 17 and 19 percentage points, respectively. It is likely that additional training would continue to improve performance.

---

> > ### Comment · Reviewer_aRW7 · 2021-08-31
> > **Response**
> >
> > Thank you to the authors for your response - updated my review with my response after "Update after responses: "

---

> > > ### Author Response · Authors · 2021-08-31
> > > **Resonse to Reviewer**
> > >
> > > We've addressed the issue with demonstrating ReLMM on other tasks (though in simulation due to time constraints), see the general response. More broadly, we would emphasize that very few prior works demonstrate automated reset-free reinforcement learning of any task in the real world, and to our knowledge, no prior paper demonstrates it for any mobile manipulation task. So while the point about the choice of task being important is well taken (and we hope that the additional task we added partially mitigates this), we believe that even with just the room cleanup task, which we demonstrate on two different sets of objects and in four different rooms, the empirical contribution of the work is significant and relevant for the CoRL community. It's not reasonable to expect the first paper on real-world automated RL for a mobile manipulation task to present a system that will work for any task that someone wants to learn, and while we acknowledge that this is a limitation of our paper (which we are explicit about), we nonetheless believe that the CoRL community will find the study of how to build a working autonomous real-world RL system to be valuable. If the community is going to make meaningful progress on RL methods that work in the real world, we need to be accepting of systems papers whose main contribution is empirical.

---

> > > > ### Comment · Reviewer_aRW7 · 2021-08-31
> > > > **Response**
> > > >
> > > > Thank you for pointing out the update with an additional task and comparison to H4ILN. To expand on my prior reply:
> > > >
> > > > I still have an issue with the lack of automated curriculum results in Table 1, since you often stress the lack of human intervention and yet the static curriculum requires this (eg in abstract "in an autonomous way without human intervention" ... "can learn continuously on a real-world platform without any environment instrumentation, without human intervention". ).
> > > >
> > > > You also state in the abstract that "Our aim is to devise a robotic reinforcement learning system for learning navigation and manipulation together, in an autonomous way without human intervention, enabling continual learning under realistic assumptions."), but much of why allows for learning autonomously comes down to the design of the task rather than the design of the system, which I think should at least be acknowledged.
> > > >
> > > > The results of H4ILN are quite helpful, as they demonstrate the far lower sample efficiency of this baseline.
> > > >
> > > > With respect to the new task, while it is helpful it is also not ideal for it to be in simulation, given your emphasis on learning in the real world.
> > > >
> > > > However, I agree that the empirical result with respect to successful learning in the real world is a notable contribution, and that some of the design decisions you chose could be applicable to later work. Given that, upon further reflection I have decided to update my recommendation to be a weak accept.

---

> > > > > ### Author Response · Authors · 2021-09-01
> > > > > **Response**
> > > > >
> > > > > Thank you for taking the time to review our work and responses. We will revise the wording to “little/minimal” human intervention with respect to the stationary curriculum while we run the autonomous curriculum in the other real world environments. The autonomous curriculum experiments don't require human intervention but are more time-consuming, however, we can now use the additional time before the final paper to add the experiments in other rooms. We will also update the abstract and introduction to specify we assume a sparse reward is either available or easy to calculate from raw sensor inputs, and where the task can be decomposed into manipulation and navigation components.

---

> ### Author Response · Authors · 2021-08-27
> **Review response continued**
>
> Q4: “The baselines being compared to are very simple, with HRL4IN being a much more suitable algorithm to compare against.”
>
> A4: HRL4IN is definitely relevant, but to our knowledge it has never been used for training in the real world. This method requires ground truth knowledge of the robot's location to evaluate the subgoal distance, while our method is designed to not require any ground truth state information except what is available directly through the robot's sensors. The need for localization makes it difficult to evaluate in the real world, and the original paper did not do so. Additionally, the HRL4IN paper reports results with 10s of millions of time steps (~10k hours of experience), while we use 40 hours. It would be impractical to run this method in the real world, though we are currently running it in simulation, and will attempt to add a simulated comparison (though training will not finish before the end of the author response period).
>
> Q5: Other references
>
> A5: We have added a discussion  Mittal et al., Li et al., Honerkamp et al., and Kindle et al. to the related work section, shown in red.

---

### Official Review · Reviewer_rFXj · 2021-07-23

**Originality:** Good
**Technical Quality:** Fair
**Clarity Of Presentation:** Fair
**Impact:** 2

**Recommendation:**

Weak Accept: I recommend accepting the paper, but will not argue for my recommendation if the majority of other reviewers have a different opinion.

**Summary:**

This paper proposes a novel reinforcement learning framework for a mobile robot to autonomously learn to perform manipulation tasks with on-board sensors only.

**Issues:**

1. Some typos: Page2, Figure 2 caption, line 4, ‘succes’ should be ‘success’; Page 7, line 307, ‘making is difficult’ probably should be ‘making it difficult’; Appendix A.2, line 516, ‘is pass’ should be ‘is passed’.
2. According to section 4.4, it’s said that in stationary curriculum, grasping policy training (Algorithm 1) is performed multiple steps when the robot base keeps being stationary, and does not mention about the navigation part, hence I initially thought that the stationary curriculum uses a stationary robot in comparison to the mobile robot used in the autonomous curriculum. But the experimental results shown in Figure 3 suggest that stationary curriculum also trains a mobile robot. So now I guess that the stationary curriculum also follows Algorithm 2 except changing the last argument from 0 to 1. Is my understanding correct? And could you please also clarify it in the paper?


**Reviewer Expertise:**

Fair: Some knowledge of the area

**Strengths And Weaknesses:**

Strengths and Weaknesses
The proposed framework requires no extra instrumentation and few human interventions. The navigation policy could be learnt to support the object manipulation tasks through the influence of grasping rewards. Experimental results show the advantage of learning both grasping and navigation policies in real world tasks.

The grasping actions are discretized to simplify the learning of grasping policy, which is quite reasonable for an initial study. But I am also very curious about applying this framework to more complicated manipulation tasks that require continuous action spaces in future studies.


**Summary Of Recommendation:**

The proposed learning framework enable the mobile robot to achieve better performance on real world tasks than other existing methods while being mostly autonomous during the learning process. It could effectively save human effort and other equipment usage, which may be significant for mobile robots operating in a large-scale environment. Hence I think that it serves as a good reference for relevant research in the future.

---

> ### Author Response · Authors · 2021-08-27
> **Review response**
>
> Thank you for your review and for pointing out the typos; we will fix these.
>
> Q1: “According to section 4.4, it’s said that in stationary curriculum, grasping policy training (Algorithm 1) is performed multiple steps when the robot base keeps being stationary, and does not mention about the navigation part, hence I initially thought that the stationary curriculum uses a stationary robot in comparison to the mobile robot used in the autonomous curriculum. But the experimental results shown in Figure 3 suggest that the stationary curriculum also trains a mobile robot. So now I guess that the stationary curriculum also follows Algorithm 2 except changing the last argument from 0 to 1. Is my understanding correct? And could you please also clarify it in the paper?”
>
> A1: The stationary curriculum is an initial phase of the training process where the robot base is stationary for 2000 grasp attempts. After the 2000 grasps, we switch to the full training process which trains both the grasping and the navigation. “ReLMM with Stationary Curriculum” means that first the stationary curriculum is used to warm up the grasp policy, and then the robots starts navigating to also train the navigation policy (and continue training the grasping policy with new grasp attempts). We have renamed the last argument to TrainGrasp to ‘st’ for “stationary” as it indicates whether the grasp training is in the stationary phase (where the robot will drop the object and keep grasping) or the joint training phase (where the robot will do a pseudo-reset and pass the reward to the navigation policy).

---

### Official Review · Reviewer_m31K · 2021-07-23

**Originality:** Fair
**Technical Quality:** Very Good
**Clarity Of Presentation:** Good
**Impact:** 2

**Recommendation:**

Weak Accept: I recommend accepting the paper, but will not argue for my recommendation if the majority of other reviewers have a different opinion.

**Summary:**

The main contribution of the paper is a robotic reinforcement learning system that learns mobile manipulation in an autonomous way in the real world, without human intervention or instrumented laboratory settings. To be more specific, the paper presents 1) a modular navigation and grasping policies with onboard sensing observations that make use of uncertainty-based exploration, 2) pseudo reset mechanism to minimize necessary human intervention, and 3) curriculum learning with two variants of grasp pre-training.


**Issues:**

Main issues to be addressed
* Better showcase the generality of the proposed method, (e.g. how can the system and the algorithm be easily adapted to new types of mobile manipulation tasks)
* Improve writing Sec 6.3 and Appendix C: clarify whether navigation reward relabeling helps and whether autonomous curriculum can be as efficient as stationary curriculum
* Real world experiment of autonomous curriculum with the proper settings: it would be great to demonstrate autonomous curriculum with proper settings (e.g. the best parameters you found in simulation) in the real world across all four environments (currently only one environment is included in Table 1)
* Add related work for RL for mobile manipulation tasks in simulation that only relies on onboard sensors, e.g. [1]. Note that [1] does not “require external instrumentation in the form of top-down camera views, oracle knowledge of object pose, or precomputed navigation maps”. The top-down occupancy map used in [1] is generated from onboard LiDAR signals.'

Miscellaneous issues & questions
* Algorithm 1: it should be G^M, instead of G^N?
* For the navigation reward, why -1 is needed from the grasping reward?
* It’s understood that the grasping problem is formulated as a contextual bandit problem (with episode length = 1). What is the episode length for the navigation problem? Is it also 1, infinite, or the steps between two pseudo resets?
* Fig 3 Left and Fig 4 are very dense and hard to read.


[1] Xia, Fei, et al. "Relmogen: Leveraging motion generation in reinforcement learning for mobile manipulation." arXiv preprint arXiv:2008.07792 (2020).


**Reviewer Expertise:**

Very good: Comprehensive knowledge of the area

**Strengths And Weaknesses:**

The paper presents a solid robotic system that supports 40 hours of real-world reinforcement learning training with minimal human intervention, which is quite impressive. The solution also has some level of algorithmic novelty that includes uncertainty-based exploration, control hierarchy of navigation and grasping, and curriculum learning. The authors extensively evaluate their solution against several baselines in a few different task setups (with and without obstacles, with and without rugs, diverse objects, varying lighting conditions, etc) and achieve favorable results compared to baselines. The authors also conduct several ablation studies in simulation for different action space and curriculum methods, and demonstrate the validity of their algorithmic design choices.

One main weakness of this work seems to be the narrow applicability of the proposed method. Although it’s mentioned the ReLMM could be used to learn other mobile manipulation tasks other than room cleaning tasks, it’s doubtful whether this is actually the case. For instance, both the pseudo-reset mechanism and overall method (Algorithm 1 and 2) seem to only work with the room-cleaning task specifically. The paper would be a lot stronger if it shows ReLMM can also be applied to a different family of tasks, e.g. pick-and-place tasks that require robot base movement. The second main weakness is the mixed results in Sec 6.3 and Appendix C and the lack of clarity in the analysis. It’s claimed that the navigation reward relabeling helps sample efficiency. Yet in Figure 4 Left, the results are very mixed, without proper analysis or hypothesis for why this is the case. Same for autonomous curriculum; It was claimed that with proper parameter tuning, autonomous curriculum can learn almost as fast as stationary curriculum (Figure 4 Center), but the opposite result is shown in Appendix C.2 and Figure 8. In general, Sec 6.3 and Appendix C has room for improvement.


**Summary Of Recommendation:**

While this paper stands as a strong system paper with real robot experiments and produces favorable results over baselines in the task of interest, there still remain questions about the generality of the proposed method and the validity of all the algorithmic design choices. A thorough response to the listed issues below will help inform the final recommendation.

---

> ### Author Response · Authors · 2021-08-27
> **Review response**
>
> Thank you for your helpful feedback. In response to the question about generality, we would like to clarify that our aim is not to propose the best possible system for solving this particular task. Rather, we aim to show how to create a real-world reinforcement learning system that enables learning mobile manipulation skills entirely from real-world interaction, with minimal human intervention. We believe that this is of interest to the CoRL community, which focuses on robotic learning and strongly encourages reporting real robot experiments. It is likely true that a method that is more proficient at grasping could be designed using DexNet and coupling learning with planning, but our claims are not about how to best solve this task, but about how to make reinforcement learning work in the real world. This is our most salient difference from the prior work discussed in the reviews, which focus on reinforcement learning in simulation where the problems of non-stationarity, sample efficiency, and sensor error are artificially removed. We have edited the introduction to make the scope of our work more clear.
>
> The basic parts of our method are not specific to object pickup, and we are currently adding a pick and place task with our method (though it will not finish training in time for the response period).
>
> Q1: “What is the episode length for the navigation problem? Is it also 1, infinite, or the steps between two pseudo resets?”
>
> A1:The navigation task is formulated as an infinite horizon problem without episode limits, however when the grasping policy takes control the __done__ signal is provided to the navigation agent that affects the Bellman backup. The state of the environment is not changed (there is no reset). These episodes are typically 5-50 timesteps long.
>
> Q2: Additional Related Work
>
> A2: Xia, Fei, et al. We reference this work as [18] in the paper. You are correct, while the work uses the shortest path to the goal for the navigation tasks, it does not use external information for the mobile manipulation. We have updated the related work in the paper to reflect this. We have also added references to Mittal et al., Li et al., Honerkamp et al., and Kindle et al.
>
> Q3: “ It’s claimed that the navigation reward relabeling helps sample efficiency. Yet in Figure 4 Left, the results are very mixed, without proper analysis or hypothesis for why this is the case. Same for autonomous curriculum; It was claimed that with proper parameter tuning, the autonomous curriculum can learn almost as fast as stationary curriculum (Figure 4 Center), but the opposite result is shown in Appendix C.2 and Figure 8. In general, Sec 6.3 and Appendix C has room for improvement.”:
>
> A3: We have updated the plots in Figure 4 to be more legible. In Figure 4, the performance potted is for the simulated room without obstacles and for a single type of object, where with the best hyperparameters the automatic curriculum can be as good as the stationary. In Figure 8 (now 9) we plot the performance for the simulated room with diverse objects (left) and with obstacles (right). We find that the relabeling reward helps significantly with the diverse objects because it encourages the navigation to go towards areas of high grasp uncertainty. However, the automatic curriculum in the obstacle room is still slower than the stationary curriculum, even with the best hyperparameters. The Figure 9 caption has been updated to explain this.
>
> Q4: “For the navigation reward, why -1 is needed from the grasping reward?”
>
> A4: For the navigation reward, we found that rewards in (-1,0) worked better than (0,1) in our simulation experiments. This, along with the discount factor, helps the robot choose shorter paths towards objects.

---

> > ### Comment · Reviewer_m31K · 2021-09-04
> > **Response**
> >
> > Thank you for the response and the additional experiments!
> >
> > I think this helps to understand the work. I'm still unsure of the general lessons to extract from it for general "real world RL" deployment. I know it comes late and it won't affect my review but it would be great to add such a section to the appendix ("General Lessons for Robot Learning in Mobile Manipulation in Real World") and really scope to what is demonstrated in the paper, e.g., it is only for pick and place tasks or tasks where there is no irrecoverable states, or any other general property of your application.
> >
> > Also, please, consider incorporating work that explores RL in real-world from Dulac-Arnold (e.g. "Challenges of real-world reinforcement learning").
> >
> > I have decided to raise my score to weak accept.
> > Independently of acceptance, please, continue this work, it is necessary to explore further how to bring RL to real-world systems.

---

> > > ### Author Response · Authors · 2021-09-04
> > > **Response**
> > >
> > > Thank you for the positive assessment of our work! We agree that adding a discussion of general lessons for robotic learning is a great idea. We unfortunately can't edit the paper at this point in the response period, but we will write a section like this to include in the final version, as we strongly agree that this would help improve the work and provide useful takeaways.
> > >
> > > We also strongly agree that scoping the claims and discussing limitations is important, so we will clarify the following points in the paper: (1) the method does not currently provide any explicit handling of safety, and this is an important direction for future work; (2) the method does not handle irrecoverable states, so tasks where objects can get stuck, broken, etc. would require some additional mechanisms to handle; (3) the current experiments focus on pick and place tasks with a relatively simple action space, and more elaborate manipulation behaviors, such as those that require coordinated application of force, would require a different control abstraction. Please let us know if there are any other important considerations for scope that would be important to add in your opinion.
> > >
> > > We'll cite the suggested Dulac-Arnold paper.

---

> ### Author Response · Authors · 2021-09-01
> **Additional experiments**
>
> Please see our additional experiments in our top level comment and in the updated supplementary, we are happy to answer any further questions.

---

### Official Review · Reviewer_W8H3 · 2021-07-27

**Originality:** Good
**Technical Quality:** Fair
**Clarity Of Presentation:** Fair
**Impact:** 3

**Recommendation:**

Weak Reject: I recommend rejecting the paper, but will not argue for my recommendation if the majority of other reviewers have a different opinion.

**Summary:**

The paper present a reinforcement learning method for training a grasping and navigation policy for a mobile manipulation robot, that learns to clean up the floor. The exploration strategy for the navigation leverages the uncertainty over the grasp success, as to explore states that will increase the probability for grasping success. The policy was trained with a physical robot in a real setting, with several training choices, like pre-training the grasping policy with automatic reset of the learning episodes and an autonomous curriculum.

**Issues:**

UPDATE: after the rebuttal I raised my score to weak reject.

- The related works are missing references to recent papers on the topic of learning mobile manipulation, like:

 https://arxiv.org/abs/2103.10534

 https://arxiv.org/pdf/2006.04271.pdf

 https://arxiv.org/pdf/2101.05325.pdf

 https://arxiv.org/pdf/2003.02637.pdf

- The preliminary section is wrong. The POMDP definition you provide is wrongly stated. The definition of the reward containing observations is wrong. There is also an abuse of notation. Moreover, in the subsequent methodology, you do not provide an explanation as to how you handle partial observability. Therefore, this preliminary section is not connected to the described methodology.

- The grasping policy is resembling the method of DexNet. Why is it important to learn these features for top-down grasping, and you do not rely on well-known and well-performing methods for top-down grasping? Moreover, you are over-simplifying the grasping problem to a simple 2D position on the thrown objects on the ground, hence computing 2D grasps, while I would argue that 6D grasps are more important for real world problems. Do you think that if you were to train a more complex grasping policy for 4D or 6D grasps, the behavior of the subsequent navigation policy would be the same? How would the higher uncertainty in orientation estimation affect the whole learning process? In algorithm 1, what is N, and pt? Please denote all notation. Finally, what is the observation? Is it RGB, depth, or point cloud? Are you employing some additional object detector to decide on the amount of objects? How is the high-level decision of the number of objects to be grasped being decided? Is this parameter already known?

- It was a bit hard for me to follow section 4.2, in the part describing the relabeling that you mention. Moreover, please state all used notation, e.g. what is $r_{n-std}$, what is n? What is weird for the navigation task, is that there is no reference to an obstacle avoidance objective. How can your navigation policy handle collisions, what are the sensorial data employed that will allow the agent to understand the difference between obstacles and objects to be grasped? In the video, it seems like the robot sometimes overrides objects on the ground. Is this a good behavior? Would this strategy be transferable to realistic mobile manipulation platforms?
On another note, you state that SAC is data efficient. Why is an algorithm that needs millions of samples to converge data efficient? This is a false statement.

- The experimental comparison is not fair. You should have compared your method to other approaches in literature that consider interactive navigation, like https://arxiv.org/pdf/2101.05325.pdf and https://arxiv.org/pdf/1910.11432.pdf
The comparison of only a random navigation strategy is not aligned with the claimed objective of showcasing advances in learning of mobile manipulation.
Moreover, I would expect an ablation on different grasping strategies, rather than a simple comparison to a scripted strategy. The plots of Fig, 3 are difficult to interpret, and I am not sure why the curves are interrupted. Also, you do not state how many times you have conducted these experiments, and what is the uncertainty, as the mean behavior is very noisy, and it doesn't look like the algorithm converged to a good behavior, e.g., in Fig. 3top, both blue and green curves are oscillating, and interrupted. There should be sufficient explanation on the presented results, and I would suggest to report success rates at least on 5 experiments with different random seeds. Additionally, the results of Fig. 4 are not allowing to derive sufficient conclusions on the algorithmic decisions.

Finally, I would recommend that the authors do not make strong claims in their conclusions that are not justified by the results, and the absence of comparisons with state-of-the-art works.





**Reviewer Expertise:**

Excellent: Expert knowledge on the topic of the paper

**Strengths And Weaknesses:**

I found the idea of the automatic resets a nice design decision for the problem of learning mobile manipulation.
A major weakness is the final behavior of the robot that seems to be pushing objects as obstacles instead of stopping and grasping them. I find the idea of proposing to solve mobile manipulation in an end to end fashion is rather greedy. Moreover, the task- oriented policies are not a new idea in robot learning, and the combination of grasp success to a subsequent policy has been previously proposed, but perhaps not in the exact same setting of this paper.

**Summary Of Recommendation:**

I have three main concerns that drive my recommendation.
1. Since top-down grasping has been generally solved, and in particular the method described resembles quite much the DexNet method, I don’t see why the authors overly analyze this part. Moreover, The objects used are very trivial, and do not correspond to objects found in a real home environment. The grasping policy is also rather simplified in an x,y position, that makes the learning rather easy.
2. Not sufficient comparisons with baselines. The authors do not refer or compare with current works on mobile manipulation, that show these benefit of coupling learning with planning, The authors also do motivate why model-free RL is a suitable method for solving mobile manipulation tasks.
3. It is not clear how the robot learns to avoid obstacles, as in the provided video it seems like the robot is sometimes pushing away objects that are fallen in the ground and should have been grasped.

---

> ### Author Response · Authors · 2021-08-27
> **Review response**
>
> Thank you for your time and feedback on our work. Our understanding of the main issues in your reviewer is that our paper does not make a strong case for why this learning-based approach is a good approach for solving the particular task in the paper. Our aim is not to propose the best possible system for solving this particular task, but rather to show how a real-world reinforcement learning system could be instantiated that enables learning a manipulation skill entirely from real-world interaction, with minimal human intervention. We believe that this is of interest to the CoRL community, which focuses on robotic learning. It's likely true that a more accurate grasping method could be designed for this particular task based on DexNet and coupling learning with planning, but our claims are not about how to best solve this task, but about how to make reinforcement learning work in the real world. We have updated the introduction to make this more clear. To our knowledge, our work is the first to demonstrate that a mobile manipulation task can be learned entirely in the real world autonomously with reinforcement learning from only first-person RGB images and proprioception.
> Comparison to (Li 2019): HRL4IN is definitely relevant, but to our knowledge it has never been used for training in the real world. This method requires ground truth knowledge of the robot's location to evaluate the subgoal distance, while our method is designed to not require any ground truth state information except what is available directly through the robot's sensors. The need for localization makes it difficult to evaluate in the real world, and the original paper did not do so. Additionally, the HRL4IN paper reports results with 10s of millions of time steps (~10k hours of experience), while we use 40 hours. It would be impractical to run this method in the real world, though we are currently running it in simulation, and will attempt to add a simulated comparison (though training will not finish before the end of the author response period).
>
> Q1: POMDP Definition:
>
> A1: We acknowledge that there were typos with our definition of the POMDP, which we have fixed in Section 3. In terms of how we handle partial observability, we learn a policy that depends only on the observation, and while this can definitely be suboptimal, we find that it works reasonably well in our case.
>
> Q2: “The grasping policy is resembling the method of DexNet. Why is it important to learn these features for top-down grasping, and you do not rely on well-known and well-performing methods for top-down grasping?”
>
> A2: While future work could incorporate methods such as DexNet into our framework, DexNet relies on large-scale simulated training. Our goal in this work was to investigate the problem of learning entirely in the real world. While our results were limited to 2D grasping, adding a greater dexterity in grasping is conceptually simple but comes at the cost of longer training time. As there already exists a very large body of work on learning to grasp, this was not the focus of our paper.
>
> Q3: Variable names:
>
> A3: In algorithm 1, N was the number of grasps to attempt, and pt (now renamed st) was an indicator about whether this is the stationary phase of the stationary curriculum. N_{st} is the number of stationary grasp attempts. In the navigation policy section,  r_n is the reward function for the navigation (in contrast to r_g for grasping); r_{n-std} was the reward function for the navigation policy with relabeling using the grasp policy’s standard deviation, which now just refer to as “relabel”. We have clarified this in the updated paper version.
>
> Q4: “What is the observation? Is it RGB, depth, or point cloud?”
>
> A4: The observation is the first person RGB image output from the LoCoBot’s camera. We updated Section 3 to include this information.
>
> Q5: “Are you employing some additional object detector to decide on the amount of objects? How is the high-level decision of the number of objects to be grasped being decided?Is this parameter already known?”
>
> A5: No, we do not employ any object detectors. At the beginning of training, we place the robot in the room along with 20 objects. This number is not given to the robot. During training, whenever the robot successfully picks an object, it executes the reset routine to place it back down. As such it can continue training forever as it does not run out of objects. During evaluation, we run the policy without the reset routine for 15 minutes (thus the robot stores them in the basket instead) and we count how many objects it failed to collect. If it were to collect all the objects the policy would continue to explore the room to try to find more until manually stopped.

---

> ### Author Response · Authors · 2021-08-27
> **review response continued**
>
> Q6: “ How can your navigation policy handle collisions, what are the sensorial data employed that will allow the agent to understand the difference between obstacles and objects to be grasped?”
>
> A6: In the last paragraph of Section 5 we described a simple collision avoidance algorithm we use for safe operation; this text is now in Appendix A.3. The navigation method is only rewarded for successful grasps, and therefore may bump other objects around. However, adding a collision penalty to the navigation reward would certainly be possible, as other works have done. This text has now been moved to Appendix A.3
> In this task, any object that can be picked up is a valid object, which is similar to other real-world grasp learning work such as [39]. However, as the robot uses its camera to determine whether a grasp was successful, a classifier could be used to discriminate between different objects.
>
> Q7: “In the video, it seems like the robot sometimes overrides objects on the ground.”
>
> A7: The robot is not explicitly restricted from colliding with obstacles or running over objects. However, our reward function will encourage the agent to learn these behaviors implicitly. The agent will receive reward earlier if the agent stops earlier and picks up these objects instead. This behavior decreases in prevalence over the course of training and could likely improve further.
>
> Q8: “On another note, you state that SAC is data-efficient. Why is an algorithm that needs millions of samples to converge data efficient? This is a false statement.”
>
> A8: SAC does not require millions of samples in our experiments. ReLMM used SAC to train its navigation policy with only 5 to 20 thousand samples. SAC can train using off-policy data that was collected in the past making it more sample efficient than many on-policy methods. That said, we have removed the statement that SAC is data-efficient, as this has nothing to do with our method or claims, and of course it may not be efficient for other problems.
>
>
> Q9: “Figure 3 is difficult to interpret.”
> A9: Due to our reset-free constraint, we cannot run the evaluation protocol regularly during real-world training (as this would require rearranging the objects). As a substitute, we plot the training performance as the number of successful grasps every 15 minutes. To compare the training in the different rooms, where the objects are more or less spaced out, we divide this number by the surface area of the room. We have clarified this in the caption. This plots is all noisy due to the non-stationary environment changes over the course of training: the lighting changes and the objects move as the robot grasps them, places them, and bumps into them. In Figure 4, we plot the simulation performance using the evaluation protocol, which you can see look more smooth.
>
> Q10: Number of times the experiments are conducted.
>
> A10: The simulation experiments are each performed with 3 random seeds, the mean and standard deviation are plotted in Figure 4. Each real training experiment is only run once per room and evaluated 3 times, with the mean and standard deviation of the evaluations listed in the table. The 4 rooms with a stationary curriculum plus 1 room with an autonomous curriculum represent 209 hours of active robot training time. Our diversity of real-world evaluations together with the simulation results show the results are significant. This information has been added to the table and Figure captions.

---

> ### Author Response · Authors · 2021-08-27
> **Review response continued (2)**
>
> Q11: Additional References:
>
> A11:Thank you for these additional references, we have added them to the updated paper in the related work section. We note that Wang et al. was already discussed in the original submission as reference [16]. For completeness, we also list here how these works differ from ours.
> Wang et al. This paper was referenced in our submission as [16]. This paper proposes training in simulation with dense trajectory tracking rewards.  Domain randomization to transfer to the real world, but requires the object state. In the real experiments, the object is located with AR markers, and only one object is present. Our work differs in many ways, including grasping multiple objects, learning from vision in the real world, and using sparse rewards.
>
> - Mittal et al. This paper focuses on learning to operate object handles. It uses a hierarchical structure, but in contrast to our work it relies on semantic segmentation of the 3D space. This segmentation model is trained on simulated objects, and the paper’s results are entirely in simulation.
>
> - Kindle et al. This paper trains a policy in simulation by combining RL and planning and transfers the policy to the real world by using LiDAR observations. Unlike our approach, this method assumes a known target end-effector position, which provides a dense supervision signal. In our work, the true object positions are never provided to the algorithm.
>
> - Honerkamp et al. This work also trains a policy in simulation by using kinematic feasibility as a reward signal. The target poses are given to the agent to execute the tasks, and in the real world, “The poses of the target objects in map frame are provided manually.” As above, our work differs by learning from image inputs and never has access to a goal in end-effector space. In our task, these poses are not available and so we cannot compare to this method without adding additional supervision such as AR tags.
>
> - Li et al. (HRL4IN). This work proposes a hierarchical policy for goal-conditioned navigation. The high-level policy is given a (x,y) goal for the robot base to reach along with its current state, and it outputs either a subgoal for the base or for the end-effector. The low-level policies aims to reach the subgoal. The hierarchical decomposition is similar to ours, but we do not use a high-level policy. Instead, the grasping policy directly takes over based on its confidence. Unlike our method, HRL4IN has not been shown to work in the real world.

---

> ### Comment · Reviewer_W8H3 · 2021-08-31
> **Rebuttal acknowledged**
>
> I thank the authors for their effort to acknowledge my doubts, and trying to deliver combative studies.
>
> My main problem lies in the contribution of the paper. As you say that "our paper does not make a strong case for why this learning-based approach is a good approach for solving the particular task in the paper. Our aim is not to propose the best possible system for solving this particular task, but rather to show how a real-world reinforcement learning system could be instantiated that enables learning a manipulation skill entirely from real-world interaction, with minimal human intervention."
>
> This is not the first time we see RL in real world setting, and for sure your restricted action space reduces the task complexity and the training time.
>
> In the end, the paper is more about how to setup an experiment in the real world, rather than addressing fundamental robotic problems that could be solved with RL in the real system. Therefore, the paper is an experimental paper that proposes ways of training reset-free.
>
> I would propose to the authors to consider rephrasing their title, and contributions to highlight this specific part of their work, e.g., setting real-world reset-free environments for practical RL, without putting a focus on mobile manipulation that you do not propose any fundamental breakthrough.
>
> Concluding, I will raise my score to weak reject.

---

> > ### Author Response · Authors · 2021-09-01
> > **Response**
> >
> > We appreciate the more positive evaluation of our work. We want to clarify two things:
> > “This is not the first time we see RL in real world setting” -- This is definitely true, and we do not claim this. What we claimed is that this is the first paper that demonstrates autonomous real-world RL for a mobile manipulation task. We believe this to be true, but if there is any prior paper we missed, we would appreciate it if you could mention it. We believe this is significant and relevant for the CoRL community, because it provides an example of how an RL system can be instantiated in an important and difficult application domain.
> > “In the end, the paper is more about how to setup an experiment in the real world, rather than addressing fundamental robotic problems that could be solved with RL in the real system” -- The paper is about how to instantiate a robotic reinforcement learning system for mobile manipulation in the real world, with an evaluation on a cleanup task (plus the additional pick and place task we added in the rebuttal). We would be happy to clarify the scope of our work in the paper if you believe some part is misleading, but our paper is not “setting up real-world reset-free environments”.
> >
> > The CoRL call for papers states that "CoRL is a selective, single-track international conference for robot learning research, spanning a broad range of topics in both theory and applications.” Our work falls under 3 categories of the CfP:
> > - “Reinforcement learning,” in that we show that using the manipulation policy’s value and uncertainty to choose when to switch between policies during training speeds up RL for a class of mobile manipulation tasks.
> > - “Applications of robot learning in robot manipulation, navigation, driving, flight, and other areas of robotics,” in that we show an application to a mobile manipulation task (cleaning up a room + additionally pick and place) and show our system working in 4 different real world environments.
> > - “Robot systems, hardware, and sensors for learning and data-driven approaches," in that we are, to our knowledge, the first to propose a robot system for learning to perform the above task in the real world without external sensors and with minimal human intervention.

---

> > > ### Comment · Reviewer_W8H3 · 2021-09-01
> > > **Response back**
> > >
> > > In my opinion there is very limited contribution, on a platform that with reduced action space. I can see the point of first training of mobile manipulation, but your method would never work in a realistic high-dof robot (10DoF) as common mobile manipulators, and I would not trust my robot to operate blindly with no obstacle avoidance, and no self-collision checking, and so on. Can you justify how this method could work on a higher action space, e.g., not doing the simplistic grasping actions?
> > >
> > > Furthermore, I do believe that the main contribution is the reset-free setting of the environment -- with a stationary curriculum, and the new results do not justify something further. Plus, I do not see what further breakthrough does the paper have to give regarding real-world deployable robots. The observation and action space is simple, hence it is reasonable to use an off-policy method and learn it. The rest of your tasks are performed in simulation - hence not real world pick and placing.
> > >
> > > And regarding the lecture about the CoRL guidelines, I am following them strictly and never said that you are out of scope. My opinion is that you provide a minimal contribution, hence I cannot recommend paper acceptance.

---

> > > > ### Author Response · Authors · 2021-09-01
> > > > **Response**
> > > >
> > > > If we are understanding your comment correctly, your main reservation appears to be with the limitations of the method and the choice of action space. For the specific question of action space, our choice is not unusual: a significant number of papers that study RL and imitation learning with real robots use a very similar representation consisting of grasp locations [1, 2, 3, 4]. We believe the contribution should be evaluated in the context of the prior work, and while we do not improve on the limitation in regard to action spaces, we do take that same representation into a new and challenging setting by adding mobility. We would be happy to clarify this limitation in our discussion, as well of course as any other limitation you believe is important -- we do want to be very clear with our readers in regard to limitations so as to enable future work to address them!
> > > >
> > > > In regard to "would never work in a realistic high-dof robot" or "would not trust my robot to operate": Perhaps the issue here has to do with the publication standard. Of course this paper does not fully solve the mobile manipulation problem, and we didn't claim that. It has limitations. We don't address safety in a way that would be adequate for large robots, we leave this for future work, and we directly acknowledge in the paper this is an important question. But no paper can do everything -- if we only publish papers that fully solve a given application problem, then we'll end up not publishing anything. Similarly, if we only publish papers that represent fundamental breakthroughs, then nothing would get published. But in the context of prior work, we believe that our contribution is significant to the community, because it demonstrates how action abstractions that were used before (see citations above) can be placed into the context of a real-world reset-free mobile manipulation RL system. This is not a breakthrough, but we believe it is a solid advance in line with the expectations for a CoRL paper. System papers are important to advance robotics -- if the bar for a system paper is to fully solve a given application, while the bar for algorithms papers is to improve on a prior algorithm, CoRL will only publish algorithms papers and no systems papers.
> > > >
> > > > As to whether a reinforcement learning system like this will be able to tackle more complex tasks, that is a good question! It seems reasonable to predict that reinforcement learning algorithms will get better, and more scalable, and therefore indeed will be able to handle high-DoF robots. Several recent papers, whose contribution is orthogonal to ours, study this question [5, 6, 7], but more generally we believe this is an important question for future work. But again, our contribution is not a better RL algorithm, but rather to study real-world autonomous RL. To make the problem manageable to study in a single paper, we have to scope our claims, and tackling 10 DoF action spaces is not in scope. Perhaps your reservation here is that you believe it is not worth working on real-world robotic RL at all, because it is not a promising direction and has too many limitations? That's a defensible perspective, but clearly a large segment of the CoRL community does consider real-world robotic RL to be important, so blocking the publication of any paper on this topic seems unreasonable.
> > > >
> > > >
> > > > - [1] Pinto and Gupta. Supersizing Self-supervision: Learning to Grasp from 50K Tries and 700 Robot Hours. https://arxiv.org/abs/1509.06825.
> > > > - [2] Song et al. Grasping in the Wild:Learning 6DoF Closed-Loop Grasping from Low-Cost Demonstrations. https://arxiv.org/abs/1912.04344.
> > > > - [3] Zeng et al. TossingBot: Learning to Throw Arbitrary Objects with Residual Physics. https://arxiv.org/abs/1903.11239.
> > > > - [4] Zeng et al. Transporter Networks: Rearranging the Visual World for Robotic Manipulation. https://arxiv.org/abs/2010.14406.
> > > > - [5] Xie et al. Iterative Reinforcement Learning Based Design of Dynamic Locomotion Skills for Cassie. https://arxiv.org/abs/1903.09537.
> > > > - [6] Akkaya I, Andrychowicz M, Chociej M, Litwin M, McGrew B, Petron A, Paino A, Plappert M, Powell G, Ribas R, Schneider J. Solving rubik's cube with a robot hand. arXiv preprint arXiv:1910.07113. 2019 Oct 16.
> > > > [7] Hwangbo, Jemin, Joonho Lee, Alexey Dosovitskiy, Dario Bellicoso, Vassilios Tsounis, Vladlen Koltun, and Marco Hutter. 2019. “Learning Agile and Dynamic Motor Skills for Legged Robots.” Science Robotics 4 (26). https://doi.org/10.1126/scirobotics.aau5872.

---

> > > > > ### Comment · Reviewer_W8H3 · 2021-09-01
> > > > > **Response**
> > > > >
> > > > > I believe the authors expose a pretty unprofessional behavior towards the reviewer. I will not be judged on any assumptions you make. Your proposed approach is a very specific, well-designed experiment to prove that you can do RL in the real world. You do not discuss any limitations and possible improvements for future real world robotic applications. Pushing forward RL for robotics, should not consider the blind execution of end-to-end learning. My only duty is to judge your paper, and I find the approach simple, and the most interesting part is the idea of not resetting, by throwing down the grasped objects.
> > > > >
> > > > > I changed my recommendation to weak reject, which means I will not oppose if the rest of the reviewers and the meta-reviewer wants to accept.

---

> > > > > > ### Author Response · Authors · 2021-09-01
> > > > > > **Response**
> > > > > >
> > > > > > > You do not discuss any limitations and possible improvements for future real world robotic applications.
> > > > > >
> > > > > > We would be more than happy to add discussion of any limitations that you believe are important! Please let us know what you consider important here, we definitely would like to discuss limitations thoroughly to enable others to build on our work, as we've stated before. We already discuss many limitations in the paper (including safety, which we agree is not addressed carefully enough for large-scale real-world robots), and would be happy to discuss them more.
> > > > > >
> > > > > > > Pushing forward RL for robotics, should not consider the blind execution of end-to-end learning.
> > > > > >
> > > > > > While this is a perspective that could (and should!) be debated in the community, we again would state that we do not agree that this is a reasonable criterion for recommending that a paper be rejected. Clearly a large segment of the CoRL community does believe that end-to-end learning is important to explore, and hence will be interested to read our paper. While we agree that there is a robust debate to be had on this topic, the review process is not the right place for that debate.
> > > > > >
> > > > > > > I find the approach simple, and the most interesting part is the idea of not resetting, by throwing down the grasped objects.
> > > > > >
> > > > > > Is having a simple approach a downside? Certainly that doesn't seem like a bad thing in general, simple solutions are better than more complex ones, all else being equal. But that said, the main contribution of the work is not any single algorithm component, but the complete system, and that combination of parts has not been presented in prior work. We show that this combination really does enable autonomous real-world RL for a mobile manipulation task, and we believe that to be significant. If parts of the solution are simple, that should be a positive thing.
> > > > > >
> > > > > > And we are sorry to hear you found our reply unprofessional -- we read carefully over the entirety of the conversation, and could not find anything that could be construed as disrespectful. We believe all of our replies were carefully thought-out and supported by evidence and citations, and we've made every effort to address each concrete issue that you've raised.

---

### Author Response · Authors · 2021-08-31
**Additional Experiments**

We’ve added new results, showing how ReLMM can be extended to learn a new task in simulation (a pick and place task that requires placing objects onto a red target rectangle), and also report on our intermediate results training HRL4IN as a baseline. We hope that these additional results address prior concerns about the generality of ReLMM, as they show it can also learn other tasks with more complex reward structures, as well as clarifying how ReLMM compares to HRL4IN.

Update on HRL4IN comparison (responding to aRW7, W8H3, and meta reviewer):

For HRL4IN, we followed the method laid out in the HRL4IN paper. To make the setup match the one in our experiments, the observation space contains the RGB camera image (instead of the depth image HRL4IN uses), global XYZ position of the robot (which is not available on the real robot and not used by our method), and the local position of the gripper. The action spaces for the high-level and low-level are the velocity of the wheels and the change in gripper XYZ position. In the same manner as ReLMM, when the gripper’s height above ground goes below some threshold, the gripper closes and picks from the ground. We use the same high-level policy reward {-1, 0} that we employ in ReLMM, where the low-level policy uses the same as the HRL4IN paper. After around 2 days of training in simulation, HRL4IN has collected a total of 1M environment steps, but the performance is still at around 0.5% objects collected on average during eval. This level of performance is expected, since the HRL4IN paper took around 30M environment steps to achieve good performance on their task. On the other hand, our method only required around 30K environment steps to learn a strong policy with 90% of objects collected on average during eval. We will continue to train HRL4IN to reach 30M steps and add the results to the final version of the paper, but we would like to emphasize again that as shown through this, HRL4IN would likely require too many samples to learn in the real world. This is not an issue with our setup -- the original paper reports 30M steps to learn. Our method is focused on real-world training, and is significantly more efficient, which we believe to be of interest to the CoRL community.

Update on generality and other tasks (responding to meta reviewer, aRW7, and m31K):

To evaluate the generality of our method to new types of tasks, we have trained ReLMM on a more challenging pick and place task, which has now been added in Appendix C. To this end, we added an additional placing policy (in addition to the picking policy), which is trained in a similar fashion as the grasping policy and uses the same action space. The placing policy is given a reward of 1 if the object is successfully placed on one of the designated areas marked red, shown in Figure 7 in Appendix C in the updated paper, and 0 otherwise. During training, we strictly follow Algorithm 2 and sample the actions according to the distribution defined in Equation 2. For further details on the experiment, including the learning graphs, see Appendix C in the updated paper. The sample efficiency is similar to ReLMM trained on the grasping tasks, considering a pick and place task takes more simulation steps to complete. Solving this additional task demonstrates that ReLMM can be extended to other tasks.

---

### Meta-Review · Area_Chair_sbfu · 2021-08-11

**Recommendation:** Accept (Poster)
**Confidence:** 3

**Metareview:**

This paper presents a real robotic system that is trained to learn navigation and grasping skills via reinforcement learning. The system is capable of learning clean-up actions based solely on onboard sensing capabilities over 40 hours of training. The components of training curricula and pseudo-resets have been appreciated by the reviewers.

Although clear and easy to follow in general, as pointed out by the reviewers, the paper presents technical inaccuracies and the methodology should be clarified.

A common issue raised by the reviewers is the deficiencies in the evaluation which seem to be set more like a semi-toy-problem that seem to  suit the method. In particular, comparison with appropriate baselines and a lack of more realistic, general tasks have been missed.

The reviewers made several comments regarding a lack of related work and appropriate references, including related work in learning mobile manipulation and hierarchical RL in mobile manipulation (HRL4IN).

Post-rebuttal ====================

Thank you for answering our questions and requests, including the addition of a baseline (HRL4IN) and for clarifying HRL4IN sample inefficiency. Overall, this paper has been well received and the reviewers have been positive about the author's response..

---

> ### Author Response · Authors · 2021-08-27
> **Meta review response continued**
>
> As shown in Figure 6 in the updated supplement, ReLMM will continue to improve its performance given more training time. Since submission, we have continued training our agents in the obstacle and obstacle+rugs rooms for 12 hours each, and found this increased their performance by 17 and 19 percentage points, respectively. These new numbers are updated in Table 1, and the training plots are in Figure 3. The video has also been updated with the improved policy at 2:20. These results show the value of our learning-based method which can continue to improve with real-world training.
>
> We hope that our methods and results will open the door to more research on learning mobile manipulation directly in the real world, rather than being limited to simulation and using privileged information such as object state. Due to our practical training requirements and cheap robot platform (LoCoBot), our system is accessible to many researchers.

---

> ### Author Response · Authors · 2021-08-27
> **Meta review response**
>
> We are happy to see that the reviewers found a number of strengths in our work. Reviewer m31k describes our “solid robotic system” as “quite impressive” and “extensively evaluated”. Reviewer rFXJ notes that our system can “effectively save human effort” and “ serves as a good reference for relevant research in the future”. Reviewer aRW7 adds that our real world experiments present “impressive success rates” and that our problem setting has only been explored “in a few prior works”. We made a number of changes that we believe address the reviewer concerns and suggestions:
> - We have clarified in the introduction our work’s aims and scope.
> - We have added additional discussion of prior work to Section 2
> - We have clarified the method and results, marked in red text in the paper.
> - We are currently running our method on a new task in simulation that requires picking and placing objects onto designated areas.
> - We are running a comparison to HLR4IN in our simulated environment.
> - We continued training 2 agents after submission and found they continued to improve with more training time. The updated performance is in Table 1 and Figure 3.
>
> In response to questions related to the task complexity, we would like to clarify that our aim is not to propose the best possible system for solving this particular task. Rather, we aim to show how to create a real-world reinforcement learning system that enables learning mobile manipulation skills entirely from real-world interaction, with minimal human intervention. We believe that this is of interest to the CoRL community, which focuses on robotic learning and strongly encourages reporting real robot experiments. To our knowledge, no previously proposed method has demonstrated real-world autonomous robotic RL of mobile manipulation tasks of this sort. It is likely true that a method that is more proficient at grasping could be designed using DexNet and coupling learning with planning (or other purpose-built components), but our claims are not about how to best solve this task, but about how to make reinforcement learning work in the real world. This is our most salient difference from the prior works discussed in the reviews, which focus on reinforcement learning in simulation. We have edited the introduction to make the scope of our work more clear. To our knowledge, our work is the first to demonstrate that a mobile manipulation task can be learned entirely in the real world autonomously with reinforcement learning from only first-person RGB images and proprioception.
>
> We do believe that our room cleaning task, trained and evaluated in 4 different room setups with different objects, is a challenge for a real-world robot to learn autonomously and entirely from scratch without supervision. Our evidence for this claim is that no prior work has demonstrated autonomous reinforcement learning of such mobile manipulation tasks. While this task might not be difficult for purpose-built methods, it does represent an advance for real-world robotic RL, and such advances are necessary to get RL to be useful. This task requires learning to see objects from various distances, controlling a robot base to navigate towards them, deciding when an object is near enough to grasp, choosing a grasp point, and finally moving towards unseen areas of the room once all nearby objects have been collected. All of these skills are learned solely from the sparse signal of grasp successes and failures.
>
> The basic parts of our method are not specific to object pickup, and we are currently adding a pick and place task with our method (though it will not finish training in time for the response period).
>
> The reviewers also pointed out several ways to improve our submission, which we have addressed to the best of our ability. We have added discussion of to Mittal et al., Li et al., Honerkamp et al., and Kindle et al. and how they differ from our work to the related work section. We have also clarified the method and variable names.
>
> Comparison to HRL4IN: HRL4IN is definitely relevant, but to our knowledge it has never been used for training in the real world. This method requires ground truth knowledge of the robot's location to evaluate the subgoal distance, while our method is designed to not require any ground truth state information except what is available directly through the robot's sensors. The need for localization makes it difficult to evaluate in the real world, and the original paper did not do so. Additionally, the HRL4IN paper reports results with 10s of millions of time steps (~10k hours of experience), while we use 40 hours. It would be impractical to run this method in the real world, though we are currently running it in simulation, and will attempt to add a simulated comparison (though training will not finish before the end of the author response period).

---

### Decision · Program_Chairs · 2021-09-13

**Decision:**

Accept (Poster)

**Comment:**

This paper presents a real robotic system that is trained to learn navigation and grasping skills via reinforcement learning. The system is capable of learning clean-up actions based solely on onboard sensing capabilities over 40 hours of training. The components of training curricula and pseudo-resets have been appreciated by the reviewers.

Although clear and easy to follow in general, as pointed out by the reviewers, the paper presents technical inaccuracies and the methodology should be clarified.

A common issue raised by the reviewers is the deficiencies in the evaluation which seem to be set more like a semi-toy-problem that seem to  suit the method. In particular, comparison with appropriate baselines and a lack of more realistic, general tasks have been missed.

The reviewers made several comments regarding a lack of related work and appropriate references, including related work in learning mobile manipulation and hierarchical RL in mobile manipulation (HRL4IN).

Post-rebuttal ====================

Thank you for answering our questions and requests, including the addition of a baseline (HRL4IN) and for clarifying HRL4IN sample inefficiency. Overall, this paper has been well received and the reviewers have been positive about the author's response..